# Purification and biochemical analysis of native AMPA receptors from three different mammalian species

Prashant Rao[1], Eric Gouaux[1,2]*

**1** Vollum Institute, Oregon Health & Science University, Portland, OR, United States of America, **2** Howard Hughes Medical Institute, Oregon Health and Science University, Portland, OR, United States of America

* gouauxe@ohsu.edu

## Abstract

The majority of fast, excitatory synaptic transmission in the central nervous system (CNS) is mediated by α-amino-3-hydroxy-5-methyl-4-isoxazolepropionic acid receptors (AMPARs), which are glutamate-activated ion channels integral to synaptic plasticity, motor coordination, learning, and memory. Native AMPARs are multiprotein assemblies comprised of a tetrameric receptor core that co-assembles with a broad range of peripheral auxiliary proteins which shape subcellular localization and signaling properties of the resulting complexes. Structure determination of AMPARs has traditionally relied on recombinant expression systems; however, these methods are not well suited to elucidate the diverse array of AMPAR assemblies that are differentially expressed in mammalian brains. While recent studies of native receptor complexes have advanced our understanding of endogenous assemblies, receptors thus far have only been isolated from rodent brain tissue. Here, we employed an immunoaffinity purification strategy to isolate native AMPARs from the brains of three different mammals–pigs, sheep, and cows. Compared to rodents, pigs, sheep, and cows are ungulate mammals, animals with closer genomic identity with humans. Here we determined the molecular size, overall yield, and purity of native AMPARs isolated from these three mammals, thereby demonstrating that structural determination and biochemical analysis is possible from a clade of mammals evolutionarily distinct from rodents.

## Introduction

AMPA receptors (AMPARs), widely regarded as the primary mediators of fast synaptic transmission in the central nervous system (CNS), are ionotropic ion channels that translate chemical signals to electrical impulses. AMPARs are cation-selective receptor assemblies enriched at postsynaptic membranes that upon activation by glutamate, elicit local membrane depolarization [1]. AMPARs are ubiquitously expressed in the CNS and thus influence many pivotal excitatory signaling pathways, most notably those that underlie synaptic plasticity, motor coordination, learning, and memory [2–4].

AMPA receptors are tetramers assembled from four homologous subunits: GluA1-4 [5]. AMPARs are organized in a three-layered architecture comprising an amino-terminal domain

**Data Availability Statement:** All relevant data are within the manuscript and its Supporting Information files.

**Funding:** This work was supported by the National Institutes of Health (NINDS) grant 2R01NS038631

to Eric Gouaux and that Eric Gouaux is an investigator with the Howard Hughes Medical Institute (HHMI). The funders had no role in study design, data collection and analysis, decision to publish, or preparation of the manuscript.

**Competing interests:** The authors have declared that no competing interests exist.

(ATD) layer, which guides functional assembly and trafficking, a ligand binding domain (LBD) layer, which harbors the glutamate binding sites, and the membrane-embedded transmembrane domain (TMD) layer, which forms the non-selective, cation permeable ion channel pore [6]. In non-desensitized states, the extracellular ATD and LBD layers are each arranged as a dimer-of-dimers with local and overall two-fold rotational symmetry, making extensive subunit interactions that play critical roles in gating and assembly [6–9]. The subunit composition of AMPARs is remarkably variable, with distinct homomeric and heteromeric receptor assemblies expressed throughout the CNS [10–12]. This architectural heterogeneity underscores functional diversity, yielding ion channels with a wide range of gating kinetics [13, 14], pharmacology [15–19], and ion channel permeation properties [20–22]. Furthermore, AMPA receptors co-assemble with auxiliary proteins which decorate the periphery of the receptor, thereby influencing the assembly [23–26], trafficking [27–30], and kinetic properties [18, 30–32] of the receptor complexes. There are over 30 AMPAR auxiliary proteins, many of which exhibit a high degree of brain region specificity, further expanding the architectural and functional complexity of AMPAR assemblies [10].

Despite decades of structural studies describing the gating and kinetic properties of recombinant AMPARs [8, 33–35], visualizing the architecture and the molecular composition of native AMPAR assemblies has proven more challenging. Indeed, while heterologous expression systems [36, 37] have succeeded in overexpressing specific AMPAR complexes [8, 38, 39], often using covalent linkage to force auxiliary subunit position and stoichiometry [9, 34, 40], isolation of native receptors from brain tissue has shed light on the diverse ensemble of receptor assemblies, not only by defining subunit composition and arrangement, but also by mapping the auxiliary subunit type and position, as well as by suggesting the presence of previously unseen auxiliary subunits [12, 41]. Moreover, the diversity of brain region- and cell-specific composition of AMPA receptor subunits and auxiliary proteins has not yet been recapitulated in heterologous expression systems. Therefore, by extracting AMPARs directly from biological tissue we can authentically define the molecular composition and architecture of AMPARs.

In a recent study, an immunoaffinity purification strategy was developed to isolate AMPARs from rat brains by leveraging the 15F1 Fab, an anti-GluA2 antibody fragment [12]. These purified GluA2-containing receptors were subjected to cryo-EM analysis, which enabled visualization of the molecular composition, subunit arrangement, and architecture of an ensemble of native AMPAR assemblies [12]. Rodent variants have been used extensively not only for structural analysis but also for electrophysiology and binding assays, providing a consistent framework for the integration of structural and functional data into mechanistic schemes. Here, we demonstrate the isolation of native AMPARs from a different clade of mammals–ungulates. Consisting of primarily hoofed mammals, ungulates have genetic and physiological traits more similar to humans than rodents [42, 43], and present advantages as models for therapeutic and translational neurological research [44–46]. Exploiting the conserved epitope of the 15F1-Fab in ungulates, we were able to isolate native AMPARs from pig, sheep, and cow brain tissue. GFP fluorescence was used to follow AMPARs throughout the isolation process to determine and compare the purity, molecular size, and total yield from ungulates and rodents, altogether revealing that structural characterization and biochemical analysis of native AMPARs is feasible from ungulate mammals.

## Results

### Rodents and ungulates share a conserved 15F1 epitope

In mammalian brains, close to 80% of AMPARs are assembled with at least one GluA2 subunit [10]. Therefore, employing the 15F1 Fab for immunoaffinity purification enables the isolation

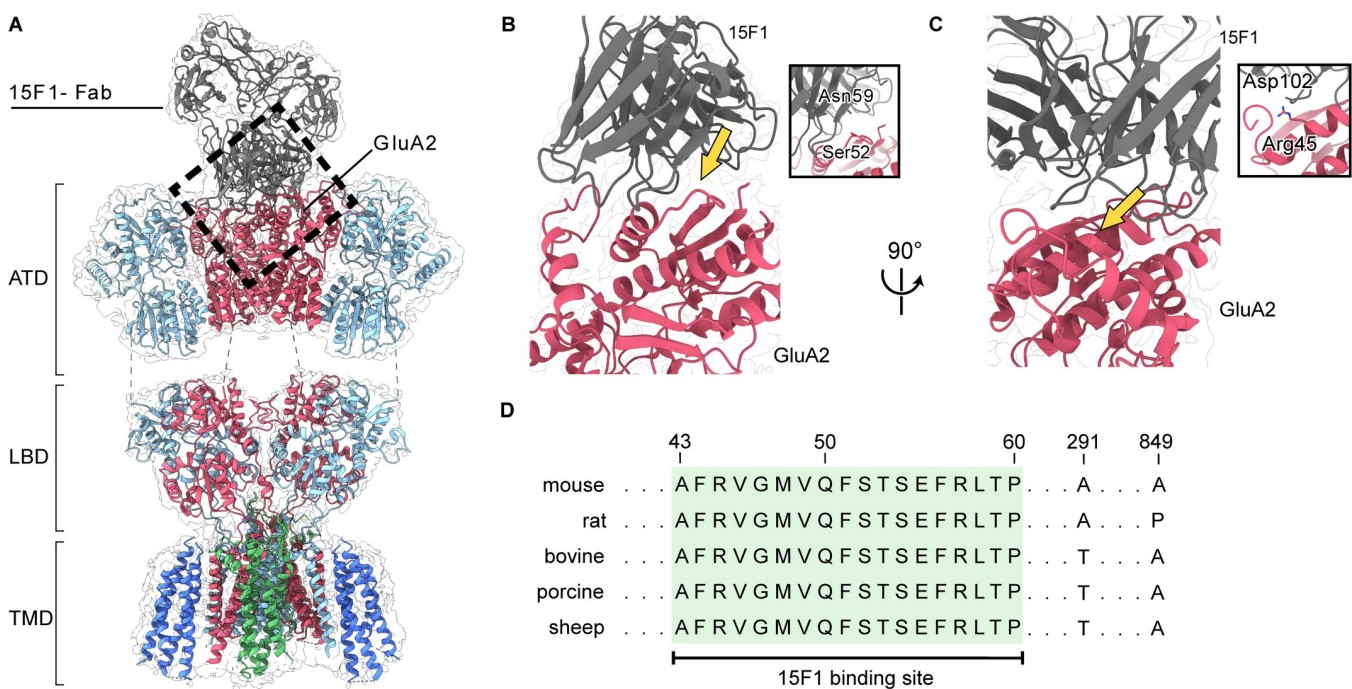

**Fig 1. Conservation of the 15F1 Fab binding site.** (**A**) Molecular model and cryo-EM map of the GluA1/GluA2 diheteromeric structure purified from mouse hippocampi (PDB: 7LDD) [41]. GluA1, GluA2, 15F1, TARP-γ8, and CNIH2 are colored cyan, red, grey, green, and blue, respectively. The black dashed rectangle highlights the 15F1-Fab binding site on the ATD layer of GluA2. (**B, C**) Views of the interaction sites between 15F1 and GluA2. Insets: Close-up views of the regions indicated by the yellow arrows. (**D**) GluA2 sequence alignment of selected mammalian species. Conserved residues of the 15F1 binding region are highlighted in green. Unshaded residues denote sequence disparities between the selected mammals.

of the major population of native AMPARs from whole brains. With rodent variants, 15F1 displays sub-nanomolar affinity for GluA2 and has no detectable cross-reactivity with the other AMPA receptor subunits [12]; however, it was unclear if these binding properties would be preserved in AMPARs from pigs, sheep, and cows. To identify the 15F1 epitope, we carefully examined the previously solved rodent AMPAR structures and found that the structural resolution of the 15F1-receptor binding region was the highest from the hippocampal GluA1/GluA2 diheteromeric complex (PDB: 7LDD) [41]. Inspection of the hippocampal GluA1/GluA2 structure permitted us to definitively map the binding region of 15F1 to the periphery of the ATD layer, encompassed by residues Ala43–Pro60, where we observe the Cα atoms from the variable loops of 15F1 within ~4–5 Å of the side chains on the upper lobe of the GluA2 ATD clamshell (Fig 1A). We denote a possible hydrogen bond interaction between Ser52 on the ATD layer with Asn59 of the 15F1 variable loop (Fig 1B), along with a possible electrostatic interaction between Arg46 of GluA2 and Asp102 of 15F1 (Fig 1C), both of which appear to be important interaction sites. However, due to insufficient resolution of the 15F1 Fab region of the density map, we cannot conclusively define these potential contacts. Nevertheless, the rodent sequences of the 15F1 epitope were compared to those of pigs, sheep, and cows, all of which exhibited complete sequence conservation (Fig 1D), indicating that an immunoaffinity purification strategy of GluA2-containing AMPAR complexes using the 15F1 Fab would likely be feasible from these mammals. Notably, the 15F1 epitope is also conserved in humans, but obtaining non-chemically treated brain tissue proved to be challenging. Before attempting our purification strategy, we modified the 15F1-Fab DNA coding sequence by cloning a GFP tag between the twin-Strep tag and the C-terminus of the heavy chain and

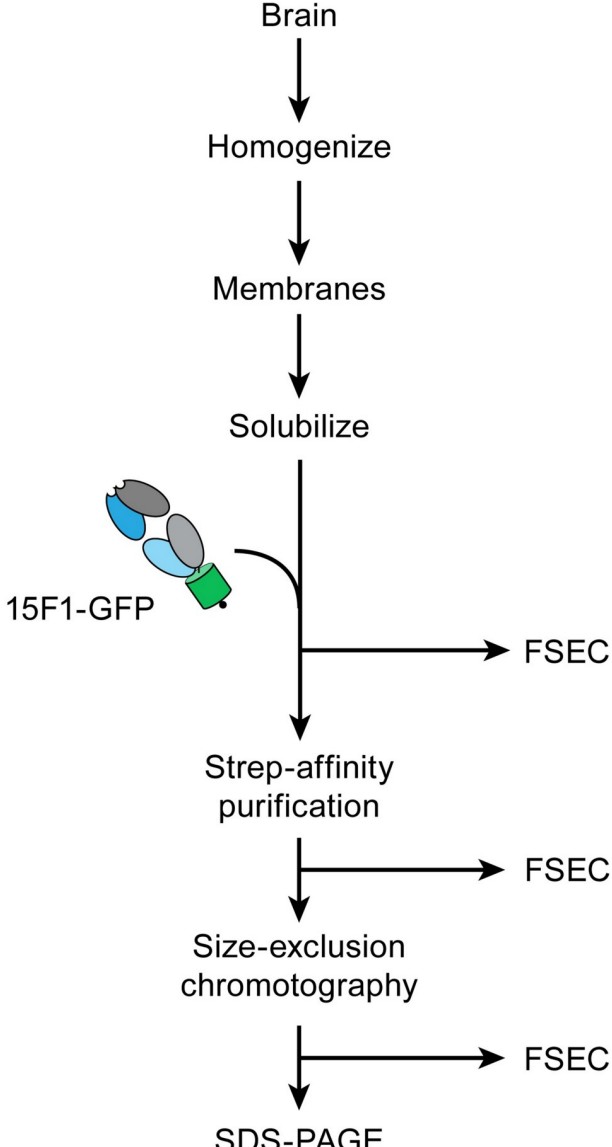

**Fig 2. Immunoaffinity purification workflow.** The outline of the immunoaffinity purification strategy. After membrane solubilization, FSEC analysis was performed at each step.

expressed this antibody in SF9 insect cells using baculovirus expression. This modification enabled us to follow AMPARs throughout the purification process using GFP fluorescence.

## Brain acquisition and membrane preparation

Using the engineered Fab, denoted 15F1-GFP, we performed identical immunoaffinity purification workflows for both rodents and ungulates (Fig 2), consistent with previous studies demonstrating isolation of native AMPARs from brain tissue [12, 41]. First, we acquired rodent and sheep brain tissue donated by neighboring researchers at OHSU. Procuring cow and pig brains proved to be more challenging, as we were unable to find any labs nearby which used these animal models. However, we inquired with commercial vendors and found that many slaughterhouses discard animal heads. Therefore, we purchased pig and cow brains from local

sources that were willing to remove whole brains from adult animal carcasses. With these whole brains in hand, we focused our efforts on brain homogenization and membrane preparation.

In light of the difference in mass of more than two orders of magnitude between rodent and ungulate brains (Table 1), we adopted two different strategies for homogenization. Prior to sonication, rodent brains were homogenized using a conventional, hand-held Dounce-homogenizer, whereas ungulate brains, due to their substantially larger mass, required more vigorous homogenization with a large blender. Post-homogenization, membranes were prepared using a two-step centrifugation strategy consisting of a low-speed 5000 x $g$ spin, followed by an ultracentrifugation step at 150,000 x $g$. The low-speed centrifugation step was implemented to first pellet insoluble tissue and cell debris. With ungulate brains, we found the majority of the brain homogenate to pellet during this step. Recovering AMPARs from this material, however, required solubilization under harsh conditions with ionic detergents, thus disrupting the native structure and likely promoting denaturation of the receptor complexes. Next, the supernatant was subjected to ultracentrifugation which pelleted the membranes. We measured the mass of membranes for each mammal (Table 1), before resuspending them in homogenization buffer.

## Detection of native AMPARs using 15F1-GFP

Solubilization of membranes using detergent is an effective approach to extract membrane proteins from the lipid bilayer [47]. We elected to use digitonin as it is capable of efficiently extracting and preserving the structural integrity of AMPARs, while also retaining co-assembled AMPAR auxiliary proteins [8, 12, 39, 41, 48]. To assess if brain size correlates with AMPAR abundance, we first solubilized equal masses of membranes from each mammal in 2% digitonin (w/v) and incubated this material with 15F1-GFP. Using fluorescence-size exclusion chromatography (FSEC) [49], we observed distinct peaks corresponding to GluA2-containing AMPARs based on GFP fluorescence and elution time (Fig 3A). We compared the peak heights of all five mammals and found that sheep and pigs exhibited the highest abundance of AMPARs, and surprisingly, we observed that cows display the lowest abundance. This discrepancy in molecular abundance was unexpected considering sheep, pigs, and cows have similar brain masses and are from the same clade of mammals. However, we are cautious not to overinterpret these FSEC profiles, as estimating abundance from solubilization of membrane pellets is challenging, due to the technical difficulty of weighing wet membranes. Nevertheless, the reason for this disparity remains unclear, although we speculate it could be due to the incidental removal of AMPARs in the first centrifugation step during membrane preparation.

Post-solubilization, we also observed that sheep AMPARs elute earlier than the other four mammalian receptor variants (Fig 3A), which we estimate corresponds to a larger molecular

**Table 1. Quantification of AMPARs and brain tissue amongst different species.**

| | Brain mass (g) | Mass of membranes (g) | Mass of purified AMPAR complexes (µg) | Receptor/brain tissue % (x $10^{-8}$) | Receptor/ membrane % (x $10^{-7}$) | Molecular weight: Solubilization (kDa) | Molecular weight: SEC-purified (kDa) | Molar quantity (picomoles) |
|---|---|---|---|---|---|---|---|---|
| Cow | 281 | 20.9 | 7.0 | 2.5 | 3.4 | 830 | 802 | 8.7 |
| Sheep | 98 | 15 | 19.7 | 20 | 13.1 | 850 | 841 | 23.4 |
| Pig | 109.8 | 4.6 | 13.7 | 12 | 29.8 | 832 | 830 | 16.5 |
| Rat | 1.7 | 0.5 | 2.1 | 124 | 42 | 827 | 811 | 2.6 |
| Mouse | 0.6 | 0.3 | 0.5 | 83 | 17 | 839 | 837 | 0.6 |

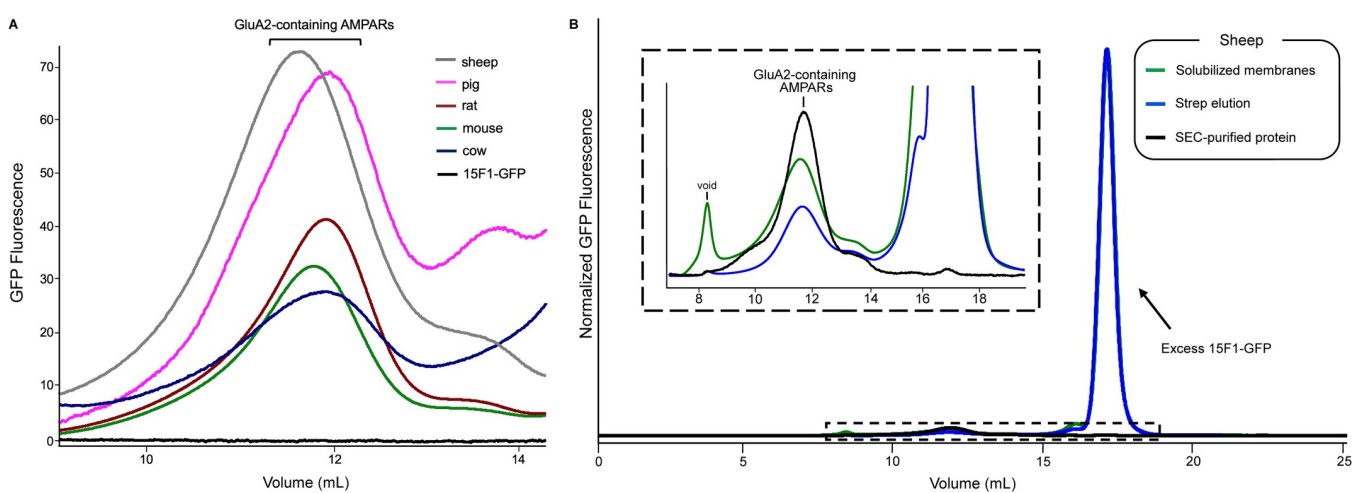

**Fig 3. Solubilization and immunoaffinity purification.** (**A**) Cross-species comparison of GluA2 elution position and abundance. The 15F1-GFP Fab was added to digitonin-solubilized membranes and the resulting samples were analyzed by FSEC. All traces were from equal masses of membranes. (**B**) FSEC traces of native sheep AMPARs analyzed throughout the immunoaffinity purification process using GFP fluorescence. Inset: Magnified view highlighting the peaks corresponding to native sheep AMPARs outlined by the dashed rectangular box.

size of 11–23 kDa (Table 1). This raises the possibility that variability in receptor and/or auxiliary subunit composition between these species account for this divergence. High-resolution proteomic analysis from rat brains has previously determined abundance profiles of auxiliary proteins that co-assemble with native AMPARs [10, 11]. The most abundant co-assembled constituents are transmembrane AMPAR regulatory proteins (TARPs), cornichons, and ferric chelate reductase 1 like protein (FRRS1L) [10], all of which bind to the receptor, largely via the transmembrane domain, and have well-described functional properties that include modulation of gating [18, 30, 32], regulation of surface trafficking [24, 26, 28, 30], and modification of pharmacology [50–52]. We compared the sequences of these auxiliary proteins from all five mammals to determine if the sheep variants have a larger predicted molecular size. The amino acid sequences were nearly identical for almost all TARPs and cornichons, although we found an additional 10, 13, 13, and 15 amino acid (aa) appendage to the *N*-terminal domain of the TARP-γ8 sheep variant compared to cows, pigs, mice, and rats, respectively (S1A Fig in S1 File). The cornichon-2 variant in cows differed from all other variants with a 15 aa N-terminal sequence addition. A comparison of the FRRS1L variants revealed a 49, 49, and 58 aa addition in the sequence of the sheep variant compared to rodents, pigs, and cows, respectively (S1B Fig in S1 File). Moreover, previous studies have shown that FRSS1L and TARP-γ8 bind in two-fold arrangements to separate AMPARs populations [24, 41], diminishing the possibility of combined assembly with the same receptor. Therefore, the amino acid extensions of both the TARP-γ8 and FRRS1L sheep variants appear insufficient to be solely responsible for the estimated differences in size.

To examine if the difference in molecular size was contingent upon the architectural variance of the receptor tetramer, we directly compared the sequences of the AMPAR subunits between all the five mammals. Sequence identity was high between receptor subunits from the different species; however, we noted several differences in predicted glycosylation sites. Sheep have more predicted O- or N-linked glycosylation sites than rodent variants for the GluA1, GluA3, and GluA4 subunits, 1, 4, and 4 more respectively. For the GluA2 subunit, the rodent variant has 1 more predicted glycan than the sheep variant. Predicting how these differences in post-translational modifications will affect the sizes of various heteromeric receptor assemblies presents a challenge, but we surmise that the variability of glycosylation contributes to the

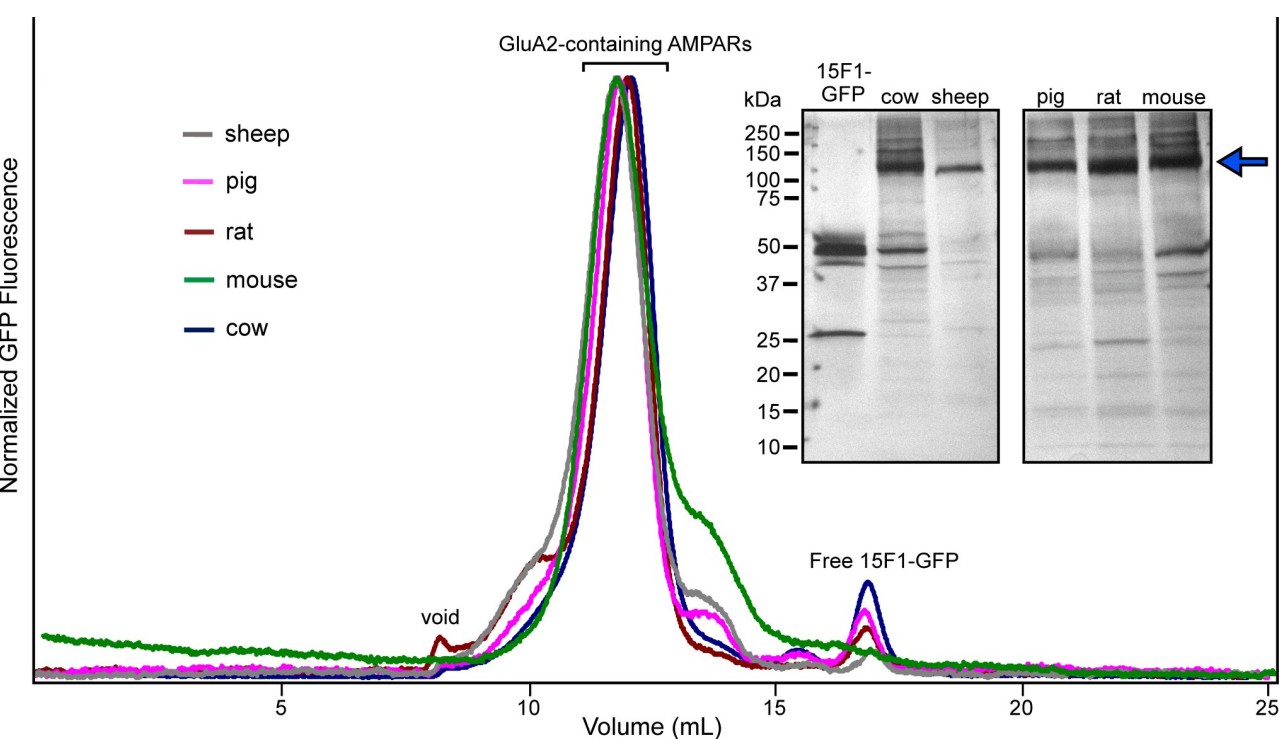

**Fig 4. Analysis of purified native AMPARs.** Overlaid FSEC profiles of 15F1 Fab-purified AMPAR assemblies from the five mammals. Inset: Silver-stained, SDS-PAGE analysis of purified receptors. The blue arrow indicates the migration position of the majority of the AMPAR subunits. The samples were run on two different gels and positioned adjacent to each other for clarity.

discrepancies in molecular size. Furthermore, we cannot rule out the possibility that differential assembly associated with auxiliary proteins may also be a contributing factor.

## Immunoaffinity purification workflow and analysis

To purify native AMPARs, we first solubilized membranes amassed from a single brain of each animal and individually incubated the digitonin-solubilized suspensions with the 15F1-GFP Fab. We immobilized 15F1-GFP-bound AMPARs to Streptactin resin using the twin StrepII tag encoded on the C-terminal region of 15F1 and released 15F1-GFP Fab by incubating the resin with d-desthiobiotin. Size-exclusion chromatography was used to separate native AMPARs from excess 15F1-GFP Fab (Fig 3B, S2 Fig in S1 File), and we determined the purity and overall yields of the different populations of native AMPARs. For all five mammals, we isolated native AMPAR complexes in the picomole (pmol) range, sufficient for biochemical, mass spectrometric, and cryo-EM analysis [53, 54]. We purified the highest amount, about 23 pmol, from sheep, and the smallest amount– 0.6 pmol from mice. We observed homogeneous FSEC profiles from all five mammals that eluted at approximately similar positions (Fig 4). SDS-PAGE analysis of the purified samples corroborated our FSEC results, as we found the most prominent bands for all species to correspond to AMPAR subunits (Fig 4). We observed additional, undetermined bands, migrating from 15–55 kDa, which we attribute to co-purified auxiliary proteins (Fig 4).

## Discussion

Strategies to purify membrane proteins from native sources have re-emerged as a subject of considerable attention in the structural biology community. The first membrane protein

structures solved from bacteria, mitochondria, and chloroplasts relied on native purification methods [55–57]. However, these strategies were limited to targets with uniquely high endogenous abundance. The advent of purification tags, along with powerful sequencing and cloning methods stimulated rapid progress in using heterologous expression methods to facilitate structural studies of a wide range of membrane proteins, and also the generation of elegant methods for interrogating their functional properties. Recently, a resurgence of improved native membrane protein purification methods has provided unprecedented insight into molecular architecture, novel protein-protein interactions, and functional mechanisms, previously unattainable by conventional recombinant methods. Isolation of native AMPARs from rodent brain tissue using the 15F1 Fab, in particular, has led to unique insights into the structural assembly and region-specific molecular pharmacology. Immunoaffinity purification and cryo-EM analysis of AMPARs from rat brains defined the stoichiometry and arrangement of a spectrum of AMPARs, including a triheteromeric GluA1/GluA2/GluA3 assembly, which was previously unknown [12]. A subsequent study used single-molecule methods to define the unique auxiliary protein stoichiometry of mouse hippocampal AMPARs and how a forebrain-specific molecule, JNJ-55511118, binds to native AMPARs and antagonizes ion channel gating [41].

In the present study, we exploited the 15F1 Fab to purify native AMPARs from cow, sheep, pig, rat, and mouse brains. Modification of the 15F1 antibody to include a covalently attached fluorophore allowed us to follow receptors throughout our purification workflow and demonstrated a commensurate strategy for isolating native AMPARs from two distinct mammalian clades. Surprisingly, we were able to purify more AMPAR assemblies per unit of brain mass from rodents, compared to the larger mammals from the ungulate clade. This purification efficiency of native rodent AMPARs was considerably higher than pigs, cows, and sheep, as the purified molar quantities from these three species were only ~15-40x and ~3-10x compared to mice and rats, respectively, even though ungulate brains have ~100x the mass of rodent brains. Whether this purification inefficiency is unique to ungulates or simply a consequence of their brain tissue composition, remains an open question. Repeating the immunoaffinity purification workflow with multiple brains from each mammal, under careful consideration of variables such as age and sex, will allow for precise interpretation of the observations we describe here. However, despite the unexpectedly lower purification efficiency, we successfully purified pmols of native AMPAR assemblies from pigs, sheep, and cow brains, a quantity sufficient for cryo-EM, biochemical, and mass spectrometric analysis. Recent studies have demonstrated the feasibility of structural studies of native membrane protein complexes [58, 59] from pmol of purified protein [58, 59].

The disparity of AMPAR abundance and molecular size between native cow and sheep AMPARs is unresolved. Whereas native pig and sheep AMPARs seem to be relatively equivalent in purity, abundance, and molecular size, native cow AMPARs display markedly lower abundance, molecular size, and purification efficiency. The possibility remains that cow brains could simply be inherently less amenable to our immunoaffinity purification strategy due to the tissue composition of their brains. Conversely, native sheep AMPARs display the highest abundance and purification efficiency compared to pig and cow AMPARs, indicating that sheep are the favorable ungulate species for structural investigations of native AMPAR assemblies.

The immunoaffinity purification methodology we outlined offers avenues for adaptation, which we presume can lead to deeper insight into native AMPAR assemblies. For example, one could apply this methodology to evaluate age-dependent associated changes of AMPARs. Prior studies in mice have suggested that distinct populations of AMPAR complexes exist in the early postnatal period, compared to adulthood [10, 60]. Resolving the architecture and

molecular composition of native AMPARs from different developmental phases offers the opportunity to shed light on the structural dynamics of AMPAR assembly. In addition, during the homogenization step we described earlier, one can introduce sedimentation and sucrose gradient centrifugation to isolate specific sub-cellular fractions such as endoplasmic reticulum (ER) membranes. AMPARs poised for anterograde trafficking at the ER have yet to be visualized and are anticipated to co-assemble with a set of architecturally distinct auxiliary proteins compared to synaptic AMPARs [24]. Furthermore, provided one has an antibody with high-affinity and specificity for a non-GluA2 AMPAR subunit, one can isolate specific heteromeric assemblies by incorporating an additional antibody affinity purification step prior to SEC. We anticipate that this purification workflow will serve as a starting point for future studies and lead to the expansion of animal models used to characterize native AMPARs.

## Materials and methods

### Ethics statement

We did not perform any experimental manipulations on live animals. We did not perform any euthanasia, anesthesia, or any animal sacrifice in this study. Pig and cow brains were obtained directly from commercial sources. Sheep and rodent brains were dissected from donated animal carcasses, euthanized prior to our acquisition.

### Expression and purification of 15F1-GFP

As previously described [41], the DNA sequences encoding the Fab domains of the light and heavy chains from the 15F1 monoclonal antibody were cloned into a bicistronic pFastBac1 vector for baculovirus expression in Sf9 insect cells, with the following modifications. The GP64 signal peptide (MVSAIVLYVLLAAAAHSAFA) was included at the N terminus of the heavy and light chains, whereas an eGFP tag, followed by a twin-Strep II tag, were introduced at the C terminus of the heavy chain. Insect cells were transduced with baculovirus and cultured at 27˚C. After 96 hrs, the supernatant was collected and the pH was adjusted to 8.0, followed by clarification at 10,000 x $g$ for 20 min at 4˚C. The supernatant was concentrated by tangential flow filtration using a 50-kDa molecular-mass cut-off filter and dialyzed against TBS buffer (20 mM Tris, pH 8.0, 150 mM NaCl) for 36 hrs. Strep-Tactin affinity chromatography was used to isolate the 15F1-GFP Fab, which was further purified by SEC in the presence of TBS buffer. Peak fractions were pooled and stored at −80˚C. Concentrated 15F1-GFP was used for purification and FSEC experiments.

### Ungulate brain tissue homogenization

Pig (age: ~1 year) and cow (age: ~2.5 years) brains were purchased directly from Tails & Trotters (Portland, OR) and Carlton Farms (Carlton, OR), respectively, and stored on ice during the drive back to the lab (15 min– 1.5 hr). Sheep (ewe, age: ~1.5 years) brains were kindly donated by Dr. Charles Roselli (OHSU). We scheduled brain dissections at least two weeks in advance, coordinating with the vendors and Dr. Roselli on a specified time to pick up brains. All ungulate brains were dissected from animal heads within 20 min and immediately placed on ice. Upon arrival to the lab, brains were washed with ice-cold PBS, before being placed in homogenization buffer (TBS + 0.8 μM aprotinin, 2 μg ml$^{-1}$ leupeptin, 2 mM pepstain A, 1 mM phenylmethylsulfonyl fluoride) with all subsequent steps performed at 4˚C. Brains were blended in an Oster blender for 2 min with homogenization buffer. The blended material was sonicated for 1.5 min, 3 sec ON, 5 sec OFF, power– 4.0, and the homogenate was centrifuged at 5000 x $g$ for 15 min. Next, the supernatant was decanted and subjected to ultracentrifugation

at 150,000 x $g$ for 1 hr to pellet the membranes. Membranes were resuspended in TBS + 1uM (R,R)-2b (N,N'-[biphenyl-4,4'-Diyldi(2r)propane-2,1-Diyl]dipropane-2-Sulfonamide) + 1 μM MPQX ([[3,4-dihydro-7-(4-morpholinyl)-2,3-dioxo-6-(trifluoromethyl)-1(2$H$)-quinoxalinyl] methyl]phosphonic acid) + 10% glycerol.

### Rodent brain tissue homogenization

From donated rat (adult Sprague Dawley) and mouse (28–40 days, C57BL/6) carcasses, whole brains were first dissected, washed with PBS, and placed carefully in ice-cold homogenization buffer. Brains were homogenized using a Teflon-glass grinder and further disrupted using a sonicator for 5 min with cycles of 3 sec on and 6 s off, at medium power, on ice. The homogenate was subjected to centrifugation at 5000 x $g$ for 15 min. The membrane fraction was collected by ultracentrifugation at 150,000 x $g$ for 1 hr at 4˚C.

### Purification of native AMPARs

Membranes from all species underwent the same purification steps unless otherwise noted. Membrane pellets were solubilized in 2% digitonin (w/v), 1 uM (R,R)-2b, 1 uM MPQX for 3 hrs with gentle agitation. Ultracentrifugation was used to clarify the material before 15F1-GFP was incubated directly with the solubilized supernatant. After incubation on ice for 20 min with gentle agitation, 15F1-GFP-bound AMPARs were column-purified with Streptactin resin with buffer A (TBS, 1 uM (R,R)-2b, 1 uM MPQX, 0.075% (w/v) digitonin) supplemented with 8 mM d-desthiobiotin and concentrated for SEC. The eluted sample was further purified using a Superose 6 10/300 column (GE Healthcare) in the presence of buffer A. Peak fractions were pooled and concentrated using a 100-kDa cut-off concentrator. The homogeneity and molecular size of the purified native AMPARs were analyzed by silver stained SDS-PAGE and FSEC.

## Supporting information

**S1 File.**
(PDF)

**S1 Raw images.**
(PDF)

## Acknowledgments

We thank Dr. Charles Roselli for donating sheep brains, Dr. John Williams and members of his lab for donating rodent carcasses, Aaron Silverman and Tails & Trotters for dissecting pig brains, T. Provitola for assistance with figures, R. Hallford for proofreading the manuscript, and the Gouaux laboratory members for helpful discussions. This article is subject to HHMI's Open Access to Publications policy. HHMI lab heads have previously granted a nonexclusive CC BY 4.0 license to the public and a sublicensable license to HHMI in their research articles. Pursuant to those licenses, the author-accepted manuscript of this article can be made freely available under a CC BY 4.0 license immediately upon publication. The content is solely the responsibility of the authors and does not necessarily represent the official views of the National Institutes of Health.

## Author Contributions

**Conceptualization:** Prashant Rao, Eric Gouaux.

**Formal analysis:** Prashant Rao, Eric Gouaux.

**Funding acquisition:** Eric Gouaux.

**Investigation:** Prashant Rao, Eric Gouaux.

**Methodology:** Prashant Rao, Eric Gouaux.

**Resources:** Eric Gouaux.

**Supervision:** Eric Gouaux.

**Validation:** Prashant Rao, Eric Gouaux.

**Visualization:** Prashant Rao.

**Writing – original draft:** Prashant Rao.

**Writing – review & editing:** Prashant Rao, Eric Gouaux.

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
