## [Decision Letter · Decision Letter 0]

17 Oct 2022

PONE-D-22-23596Purification and biochemical analysis of native AMPA receptors from three different mammalian speciesPLOS ONE

Dear Dr. Gouaux,

Thank you for submitting your manuscript to PLOS ONE. After careful consideration, we feel that it has merit but does not fully meet PLOS ONE’s publication criteria as it currently stands. Therefore, we invite you to submit a revised version of the manuscript that addresses the points raised during the review process.

Please address reviewer's concerns and submit the revised manuscript.

We look forward to receiving your revised manuscript.

Kind regards,

Janesh Kumar, Oh.D.

Academic Editor

PLOS ONE

Journal Requirements:

2. Please expand the acronyms “NIH" and "HHMI” (as indicated in your financial disclosure) so that it states the name of your funders in full.

Reviewers' comments:

Reviewer's Responses to Questions

**Comments to the Author**

1. Is the manuscript technically sound, and do the data support the conclusions?

Reviewer #1: Yes

Reviewer #2: Yes

2. Has the statistical analysis been performed appropriately and rigorously? 

Reviewer #1: N/A

Reviewer #2: N/A

3. Have the authors made all data underlying the findings in their manuscript fully available?

Reviewer #1: Yes

Reviewer #2: Yes

4. Is the manuscript presented in an intelligible fashion and written in standard English?

Reviewer #1: Yes

Reviewer #2: Yes

5. Review Comments to the Author

Reviewer #1: This paper reports for the first time, purification of AMPA receptors from the brains of three ungulate species using an affinity antibody raised against the rat GluA2 subunit. The work is technically sound, and of high quality. Traditionally, before the development of recombinant DNA technology, many biochemical studies were performed using native proteins purified from high abundance and easily harvested tissues. The ability to do this for non-rodent mammalian brains would be a major advance, and this paper reports important experiments to test the feasibility of this approach.

I have no major comments, but suggest that for clarity and accuracy the following points be addressed.

Lines 80-81: Obviously AMPARs do fold in heterologous recombinant expression systems since they generate functional ion channel complexes. What is the basis for the statement:

“Recombinant expression systems lack the equivalent post-translational folding

… essential to the biogenesis of native AMPAR assemblies.

Lines 82-84: some would argue that this is an overstatement, given that multiple structures for AMPAR Tarp complexes, likely the major species in some areas of the brain, have been solved.

Line 110: the word entire is not accurate; the remaining 20% play key functional roles in subpopulations of neurons and glia; perhaps change “almost the entire population” to ‘the major population’

Lines 169-179 are hard to follow and need revision. “Post-homogenization, membranes were prepared using a dual-step centrifugation strategy consisting of a low-speed 5000 x g spin, followed by an ultracentrifugation step at 150,000 x g. We decanted the supernatant after the first centrifugation step, discarding the pellet which consisted of insoluble tissue and cell debris. For ungulates, we found the majority of brain mass to pellet during this step (Table 1). Recovering AMPARs from this material, however, required solubilization under harsh conditions with ionic detergents.

Table 1 does not report the mass of the low speed pellet, which from my reading is what is being discussed here. Perhaps the value is inferred from the difference of the membrane pellet mass compared to the starting material? This could be clarified.

Lines 211-213 and lines 265-268 and lines 325-326. Revise. Inspection of Table 1 reveals a more complex situation with mouse (837) and sheep (841) having essentially identical masses calculated from elution times for the SEC purified proteins. Have the authors considered attempting to measure shifts in mass following deglycosylation across species? How valid is that argument that mouse and sheep AMPARs are different?

Lines 240-241 would be easier to follow if revised as follows:

Sheep have more predicted O- or N-linked glycosylation sites than rodent variants for the GluA1, GluA3, and GluA4 subunits, 1, 4, and 4 more respectively

Line 324: A likely difference between rodent and ungulate brains is that the latter have more myelinated tissue due to the larger distances between brain substructures. It is possible that sub dissection, of e.g. hippocampal or cortical gray matter tissue might give preparation that solubilize with higher efficacy.

Figure1D: is the diverse sequence triplet at position 764 due to alternative splicing of the flip/flop exon? If so the comparison should be done using the splice variant.

Figure S2. This would be much easier to follow if the same color scheme was used across all species, instead of changing the color of the SEC purified protein in panels A-D, which already contain the species identity in the inset.

Reviewer #2: This is a straightforward manuscript describing an innovative multi-species immunoaffinity purification method in ungulates. It’s impressive that the authors could attract high-purity complexes from 3 new animal species with the same rat GluA2 antibody fragment, adapted with GFP.

I have only a few suggestions where the authors could perhaps expand their descriptions and give more background information.

1)

The age of the rodent brains is described, but no information is given for the ungulate brains that I could find. Can the authors provide at least a range (or a “greater than”). I understand that the exact age of slaughtered animals is likely unclear. Could this be related to the large amount of insoluble material? Could it be related to the difference across species (were the brains possibly different ages in a meaningful way?)

2)

Usually when working with living brain tissue, the time to dissect is critical. The faster that the brain can be removed and chilled, the better. Can the authors provide some rough guidelines as to the timelines for each dissection? Probably those done on slaughtered animals were done more slowly. We do have some timescales for the transfer of the brains to the lab, but it would be nice to have a bit more detail, and even discuss in the text whether this is a valid point or, alternatively, not systematic in any way.

3)

Some perhaps naive questions on Figure 4:

a. what are the side peaks (ie at 10 and 14 ml) in the FSEC?

b. what are the higher MW bands in the silver gel?

c. The fluorescence is normalised, can you note that on the y-axis?

MInor points

——————

I don’t want to pick nits from the nice introduction - but how “unique” are glutamate receptors - there are related families, like the odorant receptors in invertebrates. I don’t know why unique is used here. Maybe the authors can adjust the text to make their insight clearer.

Around line 263: What weights are expected for the co-purified auxiliary proteins? Can you add a note in order to be explicit?

Does fig1 have swapped colours for A1 vs A2 compared to its legend?

line 75 “visualising — have” should be has?

line 285 “Recently, a resurgence of evolved native membrane protein purification methods” do you mean refined or improved? They have not been evolved in any formal sense, have they?

6. PLOS authors have the option to publish the peer review history of their article (what does this mean?). If published, this will include your full peer review and any attached files.

Reviewer #1: **Yes: **Mark L Mayer

Reviewer #2: No

---

## [Author Response · Author response to Decision Letter 0]

28 Nov 2022

Reviewers' comments:

Reviewer #1 

Comment: Lines 80-81: Obviously AMPARs do fold in heterologous recombinant expression systems since they generate functional ion channel complexes. What is the basis for the statement:

“Recombinant expression systems lack the equivalent post-translational folding

… essential to the biogenesis of native AMPAR assemblies.”

Reply: We appreciate this comment and what we mean to state is that while heterologously expressed receptors have proven valuable, and surely are functional, isolation and study of native receptors provides one with a new window through which to view receptor subunit stoichiometry and arrangement, as well as auxiliary subunit type, number and position. We have revised this portion of the manuscript to read as follows.

“Indeed, while heterologous expression systems [36,37] have succeeded in overexpressing specific AMPAR complexes [8,38,39], often using covalent linkage to force auxiliary subunit position and stoichiometry [9,34,40], isolation of native receptors from brain tissue has shed light on the diverse ensemble of receptor assemblies, not only by defining subunit composition and arrangement, but also by mapping the auxiliary subunit type and position, as well as suggesting the presence of previously unseen auxiliary subunits [12,41]. Moreover, the diversity of brain region- and cell-specific composition of AMPA receptor subunits and auxiliary proteins has not yet been recapitulated in heterologous expression systems. Therefore, by extracting AMPARs directly from biological tissue we can authentically define the molecular composition and architecture of AMPARs.”

Comment: Lines 82-84: some would argue that this is an overstatement, given that multiple structures for AMPAR Tarp complexes, likely the major species in some areas of the brain, have been solved.

Reply: We appreciate comment and have modified the text. Our intent was only to convey that recombinant expression of AMPAR complexes within a heterologous cell system cannot reproduce the heterogenous expression and abundance of AMPAR complexes, which exhibit a high degree of receptor subunit and auxiliary subunit composition based on cell type and/or brain region.

Please see revised text, lines 86-96.

Comment: the word entire is not accurate; the remaining 20% play key functional roles in subpopulations of neurons and glia; perhaps change “almost the entire population” to ‘the major population’

Reply: We have changed this part of the text to say “the major population,” as requested.

Comment: Lines 169-179 are hard to follow and need revision. “Post-homogenization, membranes were prepared using a dual-step centrifugation strategy consisting of a low-speed 5000 x g spin, followed by an ultracentrifugation step at 150,000 x g. We decanted the supernatant after the first centrifugation step, discarding the pellet which consisted of insoluble tissue and cell debris. For ungulates, we found the majority of brain mass to pellet during this step (Table 1). Recovering AMPARs from this material, however, required solubilization under harsh conditions with ionic detergents.

Table 1 does not report the mass of the low speed pellet, which from my reading is what is being discussed here. Perhaps the value is inferred from the difference of the membrane pellet mass compared to the starting material? This could be clarified.

Reply: We thank the reviewer for these comments and have appropriately rephrased the text to provide more clarity. As the reviewer noted, the masses of the low-speed pellets are not explicitly stated in Table 1, but instead, inferred based on the differences in masses of the membrane pellet compared to the starting material. 

Revised lines 169-179 in the main text: “Post-homogenization, membranes were prepared using a two-step centrifugation strategy consisting of a low-speed 5000 x g spin, followed by an ultracentrifugation step at 150,000 x g. The low-speed centrifugation step was implemented to pellet insoluble tissue and cell debris. With ungulate brains, we found the majority of the brain homogenate to pellet during this step. Recovering AMPARs from this material, however, required solubilization under harsh conditions with ionic detergents, likely promoting denaturation of the receptor complexes. Thus, the supernatant was subjected to ultracentrifugation, pelleting cellular membranes. We measured the mass of membranes for each mammal (Table 1) before resuspending them in homogenization buffer.”

Comment: Lines 211-213 and lines 265-268 and lines 325-326. Revise. Inspection of Table 1 reveals a more complex situation with mouse (837) and sheep (841) having essentially identical masses calculated from elution times for the SEC purified proteins. Have the authors considered attempting to measure shifts in mass following deglycosylation across species? How valid is that argument that mouse and sheep AMPARs are different?

Reply: We thank the reviewer for bringing up these important points. We have not performed deglycosylation experiments. As such, we cannot argue that mouse and sheep AMPARs are different. Therefore, we have modified the requested lines, accordingly. 

Revised lines 211-213: “Post-solubilization, we also observed that sheep AMPARs elute earlier than the other four mammalian receptor variants (Fig 3a), which we estimate corresponds to a larger molecular size of 11-23 kDa (Table 1).”

Lines 265-268 were removed. 

Revised lines 325-326: “Conversely, native sheep AMPARs display the highest abundance and purification efficiency compared to pig and cow AMPARs, indicating that sheep are a favorable ungulate species for structural investigation of native AMPAR assemblies.

Comment: Lines 240-241 would be easier to follow if revised as follows:

Sheep have more predicted O- or N-linked glycosylation sites than rodent variants for the GluA1, GluA3, and GluA4 subunits, 1, 4, and 4 more respectively

Reply: We have revised the sentence accordingly. 

Comment: Line 324: A likely difference between rodent and ungulate brains is that the latter have more myelinated tissue due to the larger distances between brain substructures. It is possible that sub dissection, of e.g. hippocampal or cortical gray matter tissue might give preparation that solubilize with higher efficacy.

Reply: We appreciate and agree with this comment – that sub-dissections are likely to solubilize with higher efficiency due to less myelinated tissue. However, at this time, we do not have sufficient evidence to make this claim.

Comment: Figure1D: is the diverse sequence triplet at position 764 due to alternative splicing of the flip/flop exon? If so the comparison should be done using the splice variant.

Reply: We have modified the figure such that all sequences are of the GluA2 flip variant. 

Comment: Figure S2. This would be much easier to follow if the same color scheme was used across all species, instead of changing the color of the SEC purified protein in panels A-D, which already contain the species identity in the inset.

Reply: We have revised the figure accordingly. 

Reviewer #2

Comment: The age of the rodent brains is described, but no information is given for the ungulate brains that I could find. Can the authors provide at least a range (or a “greater than”). I understand that the exact age of slaughtered animals is likely unclear. Could this be related to the large amount of insoluble material? Could it be related to the difference across species (were the brains possibly different ages in a meaningful way?)

Reply: We thank the reviewer and appreciate these comments. We have included more specificity as to the ages of the ungulates. 

Included in the methods section:

Cows: (age: ~2.5 years)

Sheep: (age: ~1.5 years)

Pigs: (age: ~1 year)

The ages of the animals could be related to the solubilization efficiency; however, we do not have any data to support this. Similarly, it is likely that the differences in solubilization efficiency is species-dependent (i.e. lifespan-dependent), however, at this time, we do not have sufficient data to make this claim. 

Comment: Usually when working with living brain tissue, the time to dissect is critical. The faster that the brain can be removed and chilled, the better. Can the authors provide some rough guidelines as to the timelines for each dissection? Probably those done on slaughtered animals were done more slowly. We do have some timescales for the transfer of the brains to the lab, but it would be nice to have a bit more detail, and even discuss in the text whether this is a valid point or, alternatively, not systematic in any way.

Reply: The time from slaughter to dissection varied between species and was not systematically performed, largely because we did not have control over the time between slaughter and removal of the brain from the animal’s skull. Therefore, we do not have an estimate of the timeline for this portion of the brain harvesting process. However, we have included a rough estimation pertaining to the timelines for brain extraction within the methods section. “We scheduled brain dissections at least two weeks in advance, coordinating with the vendors and Dr. Roselli on a specified time to pick up brains. All ungulate brains were dissected from animal heads within 20 min and immediately placed on ice.” 

Comment: Some perhaps naive questions on Figure 4:

a. what are the side peaks (ie at 10 and 14 ml) in the FSEC?

b. what are the higher MW bands in the silver gel?

c. The fluorescence is normalised, can you note that on the y-axis?

Reply: We thank the reviewer for these questions and have addressed each of them below. 

a. The molecular sizes of the species corresponding to the side peaks at 10 and 14 mL are estimated to be 1060 kDa and 570 kDa, respectively. Thus, we surmise that the peak at 10 mL represents a small population of AMPAR complexes bound to large protein complexes, perhaps derived from the PSD. The side peak at 14 mL is likely comprised of intracellular immature AMPARs, which have not formed functional receptor complexes.

b. The entities corresponding to the higher MW bands in the gel are unknown and thus we did not comment on them. They are not AMPARs and perhaps represent minor populations of co-purified contaminants, which are present in all of our immunoaffinity purifications. 

c. We have revised the y-axis labeling to note these are normalized traces. 

Comment: I don’t want to pick nits from the nice introduction - but how “unique” are glutamate receptors - there are related families, like the odorant receptors in invertebrates. I don’t know why unique is used here. Maybe the authors can adjust the text to make their insight clearer.

Reply: We have removed the word “unique” from line 47. 

Comment: Around line 263: What weights are expected for the co-purified auxiliary proteins? Can you add a note in order to be explicit?

Reply: We appreciate these questions and have modified the text to specify the expected migration range of the co-purified auxiliary proteins. 

Revised lines 211-213: We observed additional, undetermined bands, migrating from 15 – 55 kDa, which we attribute to primarily co-purified auxiliary proteins (Fig 4).

Comment: Does fig1 have swapped colours for A1 vs A2 compared to its legend?

Reply: No, the colors are correct. 

Comment: line 75 “visualising — have” should be has?

Reply: Yes, we replaced the word “have” with “has.”

Comment: line 285 “Recently, a resurgence of evolved native membrane protein purification methods” do you mean refined or improved? They have not been evolved in any formal sense, have they?

Reply: We have replaced the word “evolved” with “improved.”

---

## [Editor Report · Decision Letter 1]

1 Dec 2022

Purification and biochemical analysis of native AMPA receptors from three different mammalian species

PONE-D-22-23596R1

Dear Dr. Gouaux,

We’re pleased to inform you that your manuscript has been judged scientifically suitable for publication and will be formally accepted for publication once it meets all outstanding technical requirements.

Kind regards,

Janesh Kumar, Oh.D.

Academic Editor

PLOS ONE

Additional Editor Comments (optional):

1) Please indicated in the FSEC graphs 3B and Fig S2 that they are normalised traces. This may also be done by modifying the figure legends.

2 ) The loading order and the labels for the un-cropped Western blot do not match. The first half is inverted with respect to main figure 4 (Western blot inset). Please rectify the same. Also, indicate that "X" labelled lanes are not included in main figure.
---

## [Editor Report · Acceptance letter]

9 Mar 2023

PONE-D-22-23596R1 

Purification and biochemical analysis of native AMPA receptors from three different mammalian species 

Dear Dr. Gouaux:

I'm pleased to inform you that your manuscript has been deemed suitable for publication in PLOS ONE. Congratulations! Your manuscript is now with our production department. 

Kind regards, 

on behalf of

Dr. Janesh Kumar 

Academic Editor

PLOS ONE